# Age-Based Genomic Screening during Childhood: Ethical and Practical Considerations in Public Health Genomics Implementation

**DOI:** 10.3390/ijns9030036

**Published:** 2023-06-27

**Authors:** Laura V. Milko, Jonathan S. Berg

**Affiliations:** Department of Genetics, University of North Carolina at Chapel Hill, 120 Mason Farm Rd., Chapel Hill, NC 27599-7264, USA; jonathan_berg@med.unc.edu

**Keywords:** genetics, newborn screening (NBS), stakeholder engagement, dissemination, implementation science, health equity

## Abstract

Genomic sequencing offers an unprecedented opportunity to detect inherited variants that are implicated in rare Mendelian disorders, yet there are many challenges to overcome before this technology can routinely be applied in the healthy population. The age-based genomic screening (ABGS) approach is a novel alternative to genome-scale sequencing at birth that aims to provide highly actionable genetic information to parents over the course of their child’s routine health care. ABGS utilizes an established metric to identify conditions with high clinical actionability and incorporates information about the age of onset and age of intervention to determine the optimal time to screen for any given condition. Ongoing partnerships with parents and providers are instrumental to the co-creation of educational resources and strategies to address potential implementation barriers. Implementation science frameworks and informative empirical data are used to evaluate strategies to establish this unique clinical application of targeted genomic sequencing. Ultimately, a pilot project conducted in primary care pediatrics clinics will assess patient and implementation outcomes, parent and provider perspectives, and the feasibility of ABGS. A validated, stakeholder-informed, and practical ABGS program will include hundreds of conditions that are actionable during infancy and childhood, setting the stage for a longitudinal implementation that can assess clinical and health economic outcomes.

## 1. Introduction

Public health newborn screening (NBS) epitomizes the early detection of individuals with rare genetic conditions to allow the initiation of effective treatments before symptoms develop. Currently, the number of detectable conditions is limited by the technology employed. Exome and genome sequencing (collectively referred to here as “genome-scale sequencing”) has profoundly advanced the ability to diagnose and manage patients affected with rare Mendelian conditions [1,2,3] and could vastly expand the range of conditions detected in the general population through predictive screening [4]. Furthermore, the promise of many cutting-edge gene and cell therapies for early onset genetic conditions is contingent on their timely ascertainment in the healthy pediatric population [5].

Proponents of expanding public health NBS to include genome-scale sequencing advocate for the predictive—and, potentially, future diagnostic—utility of providing all newborns with a sequenced genome [6]. However, despite some interest among a subset of the population, there are well-founded doubts about whether and how genome-scale sequencing should be implemented [7,8,9,10]. It is clear that the addition of genome-scale sequencing to routine NBS would raise substantial ethical, societal, and practical concerns related to parental consent and in determining the kind of information that should be sought and disclosed [11,12,13], making its implementation in NBS programs untenable at present. However, if costs are low enough and the value is high enough, predictive genomic analysis could become accessible in a way that reduces barriers to obtaining highly actionable information with the greatest chance of societal benefit.

We suggest that focusing on genome-scale sequencing at birth overlooks a crucial opportunity to deliver actionable information closer to the point of care across the lifespan. Beyond newborn screening, routine pediatric preventive care interventions continue throughout childhood, such as growth charting, vaccinations, and hearing/vision screening, all of which have individual utility and are also widely accepted because of their public health impact. The periodicity schedule for these interventions is reviewed in an evidence-based manner [14] and establishes policies for insurance coverage under the Affordable Care Act. Practitioners are accustomed to standard workflows, and parents expect to receive anticipatory guidance and screens for potential health concerns for their child over time. Given the rapid development of next-generation sequencing capabilities, could population-based genomic screening for monogenic conditions that are actionable in childhood be incorporated into the existing practice of well-child care?

An age-based genomic screening (ABGS) approach seeks to deliver discrete, predictive genomic information at intervention-oriented time-points during well-child care, including neonatally, and provide clinically useful results in easily digestible portions for both parents and providers. Thus, ABGS addresses many, but not all, of the ethical concerns about genomic sequencing in children, while retaining most of the benefits. By interrogating panels of highly actionable conditions at relevant time points during infancy and childhood [15], the focused approach of ABGS simplifies parental decision-making around the types of genetic information that should be sought and disclosed in a healthy child by screening only for conditions with the highest degree of clinical actionability at the most clinically relevant intervals. As an alternative to screening newborns for every detectable condition at birth, including those that will likely not manifest until later in childhood, ABGS ameliorates loss to the follow-up of individuals for whom there is a protracted latency between the discovery of the genetic diagnosis and the actionability of that information. Periodic engagement about genomic screening between providers and parents (and gradual exposure of the child over time) has the added benefit of building knowledge about genetics and confidence in the ability to make recommendations or decisions, thus improving genomic literacy. If implemented throughout the lifespan, ABGS could represent a cost-effective and tractable way to capitalize on advances in genomic medicine that would seamlessly connect with genomic screening programs designed for adults [16,17].

The implementation of ABGS as a public health care service, offered in diverse populations with varying geographic, sociodemographic, and cultural backgrounds will require ongoing partnerships in clinics and communities to ensure broad accessibility to parents across all segments of society. Frameworks, theories, and models from implementation science can be used to identify potential barriers and strategies to overcome them, as well as harmonized measures developed by genomic medicine consortia [18,19,20,21], which can be used to standardize data collection, knowledge integration, and inform best practices for future implementation trials [22].

The lag in the clinical adoption of genomic interventions, including population screening, into routine clinical practice has been attributed to various factors related to the impending shift from genetic specialty providers to primary care providers, including contextual factors (e.g., a perceived lack of knowledge, expertise, and supporting resources) and process factors (e.g., an engagement of organizational leadership and provider buy-in) [22,23,24]. The development of educational and clinical decision support tools for non-genetics clinicians is growing; however, research to develop and evaluate strategies to effectively integrate genomic screening into diverse clinical settings is largely unexplored, particularly in settings with limited resources and other structural barriers to implementing public health innovations [25,26]. Engagement with clinical providers and staff will be needed to develop and provide resources to support the integration of ABGS into the clinical workflow. Key stakeholders from pediatric clinics in diverse communities will need to be involved, including the perspectives of providers and parents to develop strategies and resources to facilitate the equitable uptake of genomic screening [27].

## 2. Identifying Appropriate Conditions for Screening during Childhood

Current NBS focuses on a small number of conditions with unambiguous health benefits (for the most part) and bypasses explicit parental informed consent due to its public health importance. Genetic sequencing would introduce a much wider range of possible findings for NBS, including conditions with less clinical actionability and later ages of onset that would likely invoke a paradigm shift to a new “opt-in” model of screening. Decisions about which conditions to include as part of a public health genomic screening program must take into account the value to society and various interested parties.

Partitioning the genome into interpretable groupings based on the clinical actionability and natural history characteristics of genetic conditions can facilitate informed decision-making and preference setting by individuals and parents [28]. ABGS will rely on a semi-quantitative metric (SQM) for scoring clinical actionability based on five key parameters (the severity and likelihood of disease manifestations, the efficacy and acceptability of interventions, and a knowledge base about the condition) [29] that was previously developed by our research group and subsequently adopted with minor modifications by the ClinGen Actionability and Pediatric Actionability Working Groups [30,31]. We utilized this approach to define a subset of conditions that qualified for disclosure to research participants as a “next-generation sequencing newborn screen” [32] and further analyzed SQM scores for a comparison of commercial panels that are being offered to parents as expanded newborn screening options [33].

Although clinical actionability is an important criterion for determining eligibility for screening purposes, it is not the only factor involved. The latency between information and action (the time that elapses between learning information about a newborn’s future health condition and the time at which actions can be taken to ameliorate adverse outcomes) will influence the efficacy of the genomic screening intervention. For example, a long delay could induce unnecessary stress on the family and/or child, or even impact the effectiveness of the intervention (for example, if the molecular diagnosis is not followed up on effectively, or forgotten, over the course of many years). The economic implications of genomic screening are also likely to be influenced by the duration and nature of interventions that are prompted by a positive finding, with greater downstream costs associated with prolonged periods of surveillance by medical specialists before symptoms manifest and/or definitive prophylactic or therapeutic interventions are deployed.

An ABGS approach will therefore categorize groups of conditions based on high clinical actionability, natural history pattern (age of symptom onset), and current recommendations for age of intervention. Clustering conditions in panels that can be offered at specific time points will optimally balance early detection with proximity to a recommended course of action to prevent or ameliorate symptoms. Sequencing panels will ideally be informed by and synchronized with existing pediatric preventive care visits to maximize efficiency for clinical workflows. There are clear trade-offs in terms of gene content, number of panels, and timing of testing, requiring multidisciplinary and multisource input and consensus.

## 3. Sequencing Healthy Newborns and Children: Start Small, Grow with Time

The generation of genome-scale data at birth is viewed by some as an efficient and cost-effective way to incorporate public health screening for a much broader range of conditions. Once sampled, sequenced, and stored, whole genome sequence data theoretically offers the ability to repeatedly interrogate the data if undiagnosed conditions with a suspected genetic etiology arise, and for additional screening once a child reaches the age of consent. However, the wide range of possible results from genome-scale sequencing, including conditions with differing levels of clinical actionability and ages of onset, immediately raises questions about what information should be sought and disclosed in healthy newborns. Parental reactions might include anxiety regarding: the range of possible choices about what information they wish to have disclosed to them; perceptions that elective genome sequencing is less important due to other factors in their lives that take precedence; concerns about out-of-pocket expense; worries about data privacy and legal implications for their child in the future, such as insurability; or uncertainty about whether to learn about health conditions that are out of their control. If genome-scale sequencing were adopted as part of routine public health NBS, these concerns would need to be addressed through comprehensive educational programs and detailed parental decision-making processes, which would require substantial changes to the current procedures. The neonatal period, which is clearly challenging for most parents, may not be an ideal time to make such complex decisions, and could result in parents opting out of NBS entirely [34]. Therefore, we suggest that initial public health genomic screening in newborns should focus narrowly on conditions that would be highly actionable in the neonatal period, with additional layers of screening throughout childhood—with parental informed consent—gradually expanding the number and range of conditions that are screened for across the lifespan.

Generation of genome-wide data also introduces many passionately argued ethical concerns that could impede the widespread adoption of genomic screening in infants and children. It is unlikely that extensive pre-test genetic counseling will be feasible for a population-level genomic screening program, whereas highly focused sequencing panels provide a streamlined opt-in choice for parents and providers. While it is possible to sequence the genome or exome and then limit the informatic analysis to subsets of genes, this approach requires justification as to why certain information is not being disclosed and may result in challenges with parental requests for the full set of raw data. The ABGS approach is to start by interrogating the small fraction of genes that do meet that standard, determine how best to implement genomic screening in diverse populations, and then scale it to the general population to obtain the greatest impact, for the most people, at the least expense.

A targeted sequencing approach would reduce technical issues and privacy concerns related to accessing, interpreting, and securely maintaining the enormous volume of data generated by genome-scale sequencing, including potential ethical questions about data interpretation and ownership that could hinder widespread acceptance. Finally, economic arguments about various sequencing methods can be fraught with assumptions, given the highly subsidized nature of large-scale sequencing in both the research and commercial settings, differences in economies of scale at different steps of the sequencing process, and the rapid and unpredictable advances in technology. Sequencing costs will invariably continue to reduce over time, making serial resequencing progressively less expensive, while taking advantage of newer technology and avoiding data storage and reanalysis costs (which could be substantial but are often ignored). A persistent desire to utilize better sequencing technologies as a person grows is to be expected, rather than relying on old sequence data generated at birth. Studying a population-level implementation of targeted screening (either through specific biochemical enrichment strategies or via informatics approaches to selectively analyze virtual panels) is a cost-effective and ethically feasible way to identify rare individuals with highly actionable conditions. However, at an individual level, this approach can also lay the groundwork for additional tiers of optional conditions that could be analyzed over time based on parental decision-making in consultation with a primary care provider or genetic specialist.

## 4. Using Implementation Science to Advance Innovations in Genomic Screening

We argue that ABGS of healthy children for monogenic conditions is initially and inherently an implementation challenge that must be researched and conducted in a rigorous, transparent, and relatively cautious manner in order to build public health buy-in and avoid “putting the cart before the horse” by overextending the reach of expanded genomic NBS and causing iatrogenic harm to parents and families. An early implementation of ABGS will require ongoing engagement with primary care providers and parents in order to develop an evidence-based and contextually appropriate program and responsive implementation strategies for the clinical adoption and inclusion of representative populations. Implementation science provides frameworks and models to guide the planning and conduct of implementation, as well as the evaluation of multilevel outcomes in serial implementation phases [35,36]. These frameworks and models are increasingly used to inform the design and evaluation of health interventions with a diverse reach to improve adoption and increase health equity in underserved and marginalized communities [37,38].

Implementation science approaches developed and evaluated by genomic medicine consortia will be used to standardize data collection, integrate knowledge, and inform best practices for future pragmatic clinical trials. The genomic medicine integrative research (GMIR) framework [39] developed by the National Human Genome Research Institute (NHGRI)-funded Implementing Genomics in Practice (IGNITE) network [40] and the Clinical Sequencing Evidence-Generating Research (CSER) consortium [21] has a strong theoretical basis in the consolidated framework for implementation research (CFIR) [41] and provides an organizational structure for key domains of the greatest relevance to genomic medicine researchers [42]. The GMIR framework provided an effective model to modify for ABGS study activities and will enable us to link our constructs and outcomes with those of other genomic medicine research studies and consortia [43,44] (Figure 1).

## 5. Multilevel Partnerships with Parents and Providers

The involvement of primary care providers and parents will be critical to identify and address barriers that could impair the successful clinical implementation of ABGS [45,46,47]. The understanding of negative/normal results [48] by parents and providers, and their perspectives about the future health of the child, need to be studied. Even true positive findings can have highly varying clinical utility depending on the condition, the age at which sequencing is performed, and the age at which actions need to be taken to prevent adverse health outcomes [49,50]. There are important ethical concerns that pre-symptomatic genomic sequencing could impair parental bonding, create “patients in waiting”, or impinge on the future autonomy of the child [51,52,53,54]. Extensive multi-level engagement and collaboration with parent and provider partners throughout the process will be crucial for developing the ABGS program and exploring key contextual factors for individuals and families, as well as providers and clinics, the psychosocial aspects of participation, and the broader outcomes for health and social policy.

Collaboration with primary care specialties (family medicine, internal medicine, and pediatrics) and a strong representation of minority populations will facilitate a broad perspective about the needs for ABGS implementation and foster an organizational culture that is receptive to this new clinical innovation. Partnerships with community clinics and health providers in diverse settings will help elucidate organizational readiness and identify local barriers and facilitators for the implementation of ABGS. Ongoing and iterative feedback from clinical staff and providers will be needed to guide the development of strategies and supports that are responsive to the needs of health providers and parent partners.

The effective and equitable integration of genomic screening into health care for infants and children will require building trust with community partners in diverse settings to understand the kinds of information that should be sought after and how best to communicate that information. One method to forge such a relationship is through community advisory boards that have an iterative role in the development of research objectives and resources and providing guidance and recommendations as research advances [27]. Bidirectional and ongoing relationships between researchers, providers, and communities can promote trust and genetic literacy, develop resources to facilitate informed decision-making by families, and work toward an equitable uptake of genomic screening [55].

## 6. Conclusions

ABGS transforms the thought process from “what information should be returned?” (if genome-scale sequencing is performed) to “what information should we seek out, and when?” (for the greatest benefit in the general population). Although divergent from the currently prevailing notion that genome-scale sequencing will be imminently adopted as part of routine clinical care, ABGS could establish a new clinical practice paradigm and ultimately form a stable foundation for a more widespread adoption of evidence-based and genome-driven health care. It is also entirely compatible with a future in which genome-scale sequencing is in fact routine, providing a framework for supporting decision-making, analysis, and result disclosure that would allow parents to obtain periodic analyses of actionable conditions that are highly relevant to their child’s developmental stage, and prepare individuals to receive sequencing results for carrier screening and actionable adult-onset conditions when they are able to assent or consent. Ultimately, our overarching goal should be to make the advances of genome sciences broadly available to the general population, maximizing the benefits and minimizing the harms.

## Figures and Tables

**Figure 1 IJNS-09-00036-f001:**
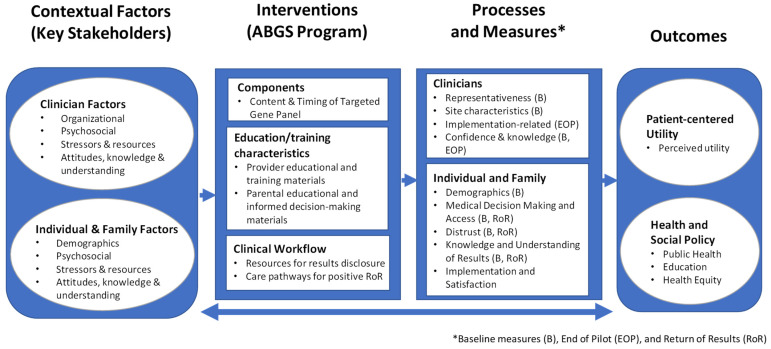
The Genomic Medicine Integrative Research (GMIR) framework developed by the National Human Genome Research Institute (NHGRI)-funded Implementing Genomics in Practice (IGNITE) network and the Clinical Sequencing Evidence-Generating Research (CSER) consortium provides an organizational structure for the key domains of the greatest relevance to genomic medicine researchers. The ABGS-adapted GMIR framework shown here will facilitate linking our constructs and outcomes with those of other genomic medicine research studies and consortia.

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
