# Peer review of "Age-Based Genomic Screening during Childhood: Ethical and Practical Considerations in Public Health Genomics Implementation"

_2409-515X, 2023, doi:10.3390/ijns9030036_

Round 1

Reviewer 1 Report

Great and interesting topic to be presented. Paper is however somewhat difficult to follow due to insufficient structure.  No clear introduction of the global problem and current possible approaches with cons and pros. The authors favor one approach without clear comparison from different angles. The commentary feels more of a "praise" of one approach. The commentary could be improved by providing a more balanced view. The implementation considerations should be described more clearly.

Author Response

We thank the reviewers for their time and effort in reviewing the paper and for their constructive feedback. We have edited the manuscript to address these comments and feel that it works better as a commentary after the revisions.

Reviewer 1:

Great and interesting topic to be presented. Paper is however somewhat difficult to follow due to insufficient structure.  No clear introduction of the global problem and current possible approaches with cons and pros. The authors favor one approach without clear comparison from different angles. The commentary feels more of a "praise" of one approach. The commentary could be improved by providing a more balanced view. 

Author response: The Introduction has been substantially revised and reorganized to provide a clearer and more balanced view of the pros and cons of different approaches.

The implementation considerations should be described more clearly.

Author response: Thank you for this comment. Indeed, we think that it is critical for novel genomic approaches like this to undergo rigorous implementation science research to ensure that they are broadly accessible. We have added text in section 4: “Using implementation science to advance innovations in genomic screening” to center the overarching implementation considerations of the ABGS approach.

Reviewer 2 Report

In this manuscript, the authors describe a system of genomic newborn screening that evaluates for different conditions across the lifespan. This is overall a rational and commendable approach to genomic newborn screening. The article would benefit from further clarity, however, regarding exactly what is happening on the genomic side, particularly as the workflow itself represents an innovative and interesting approach to genomic healthcare. It would also be helpful to lay out more clearly from the outset whether this article is presenting a study protocol or serving as a commentary on this issue.   Abstract:    Typographical error with "imple-mentation" occurring twice.   It is unclear whether the ABGS approach incorporates targeted sequencing or a genome-wide approach with analysis of targeted genes. It is also unclear whether this article is describing a study protocol.   Manuscript I think this description of the approach to variable screening over the lifespan is in need of a more detailed explanation of the genomic workflow. Particularly,  it is unclear what "genome-scale sequencing" means - would be helpful to use either a more commonly-seen term such as "genome-wide sequencing" or clearly define what this means (genome sequencing? large gene panels? analysis of a subset of genes on an exome backbone?). Later on, this is taken to refer to exome or genome sequencing, although again mentions targeted sequencing of panels (line 59, page 2) in terms of restricting evaluation to certain conditions. As it initially seems like the authors are referring to sequencing of the exome or genome with targeted analysis for certain panels of genes, but later on seems to mean repetitive panel sequencing over the years, it would be helpful to be consistent throughout regarding the extent of both the sequencing and analytic approach.     This has important implications for the implementation approach as well as for the acceptability to providers and patients, economics, potential for disparities, etc. While repetitive sequencing has the advantage of applying improved techniques, if this is also accompanied by repetitive blood draws, test orders, etc, this would present serious workflow considerations. The advantages and drawbacks of a genome up front with repeated analysis for different genes, versus repetitive sequencing over time, would be helpful to at least mention.    Finally, it is a bit unclear whether this article is a commentary on potential future applications of genomic screening, or presenting a study protocol to evaluate this approach. Additional clarity on this would be helpful, particularly as the authors reference a study they are conducting in this area. If this is the presentation of a study protocol, that should be directly stated in the abstract, and additional detail on the design and methodology should be presented.

Author Response

We thank the reviewers for their time and effort in reviewing the paper and for their constructive feedback. We have edited the manuscript to address these comments and feel that it works better as a commentary after the revisions.

Reviewer 2:

In this manuscript, the authors describe a system of genomic newborn screening that evaluates for different conditions across the lifespan. This is overall a rational and commendable approach to genomic newborn screening.  The article would benefit from further clarity, however, regarding exactly what is happening on the genomic side, particularly as the workflow itself represents an innovative and interesting approach to genomic healthcare. It would also be helpful to lay out more clearly from the outset whether this article is presenting a study protocol or serving as a commentary on this issue. 

Author response: Though we are conducting an ongoing study of ABGS, this article is intended to present a commentary on the ABGS approach and not a study protocol.  We have revised our language to refer to the approach instead of the study to clarify the intent of the paper.

Abstract:    Typographical error with "imple-mentation" occurring twice.

Author response: We fixed this journal formatting issue in the current version, will report to editors if it persists.

It is unclear whether the ABGS approach incorporates targeted sequencing or a genome-wide approach with analysis of targeted genes. 

Author response: The ABGS approach advocates for targeted sequencing panels during the initial implementation of a public health DNA-based screening in a healthy pediatric population and have made the rationale more clear in section 3: “Sequencing healthy newborns and children: start small, grow with time”.

I think this description of the approach to variable screening over the lifespan is in need of a more detailed explanation of the genomic workflow. Particularly,  it is unclear what "genome-scale sequencing" means - would be helpful to use either a more commonly-seen term such as "genome-wide sequencing" or clearly define what this means (genome sequencing? large gene panels? analysis of a subset of genes on an exome backbone?). Later on, this is taken to refer to exome or genome sequencing, although again mentions targeted sequencing of panels (line 59, page 2) in terms of restricting evaluation to certain conditions. As it initially seems like the authors are referring to sequencing of the exome or genome with targeted analysis for certain panels of genes, but later on seems to mean repetitive panel sequencing over the years, it would be helpful to be consistent throughout regarding the extent of both the sequencing and analytic approach. This has important implications for the implementation approach as well as for the acceptability to providers and patients, economics, potential for disparities, etc. While repetitive sequencing has the advantage of applying improved techniques, if this is also accompanied by repetitive blood draws, test orders, etc, this would present serious workflow considerations. The advantages and drawbacks of a genome up front with repeated analysis for different genes, versus repetitive sequencing over time, would be helpful to at least mention.

Author response: We appreciate this feedback and recognize the challenge of presenting a new proposal like ABGS in this format without space to provide full technical details.  That being said, we define our use of “genome-scale sequencing” in the Introduction and maintain consistency throughout the article. The contrast between sequencing, analyzing, and reporting on the entire genome or exome at birth (“genome-scale sequencing”), versus analyzing and reporting on focused panels of highly actionable conditions across the lifespan (“targeted sequencing”), is the key comparison we are trying to make.  The actual technical details of the sequencing method (targeted enrichment prior to sequencing, versus comprehensive sequencing followed by targeted analysis) are a nuanced discussion and will depend on cost structures and economies of scale that we recognize will change over time as the technology evolves.  Therefore, we are focused on the concept of “what should be sought out and disclosed in a public health setting” when considering the targeted/focused panels of genes that would be offered to parents in the ABGS program.  We have added significant text to section 3 and other passages throughout to define and support our approach, and have provided some additional framing of the larger debate and our arguments in favor of a limited approach that can “start small” and “grow with time”.

Finally, it is a bit unclear whether this article is a commentary on potential future applications of genomic screening, or presenting a study protocol to evaluate this approach. Additional clarity on this would be helpful, particularly as the authors reference a study they are conducting in this area. If this is the presentation of a study protocol, that should be directly stated in the abstract, and additional detail on the design and methodology should be presented.

Author response: This article is intended to be an opinion about the ABGS approach and not a detailed study protocol.  In response to this comment, we now refer to the “approach” instead of the “study” throughout.

Round 2

Reviewer 1 Report

We feel the paper can now be accepted for publication 

Reviewer 2 Report

The authors have adequately addressed my questions and comments.